# Precision improvement for indoor positioning based on fuzzy inference system with ultra-wideband wireless communications

**Shu-Hung Lee¹, Shu-Wai Chang², Yung-Fa Huang 🔾²\*, Yung-Hoh Sheu³\***

**1** School of Intelligent Manufacturing and Automotive Engineering, Guangdong University of Business and Technology, Zhaoqing, China, **2** Department of Information and Communication Engineering, Chaoyang University of Technology, Taichung, Taiwan, **3** Department of Computer Science and Information Engineering, National Formosa University, Yunlin County, Taiwan

\* yfahuang@cyut.edu.tw (YFH); yhsheu@nfu.edu.tw (YHS)

## Abstract

This paper investigates the enhancement of positioning accuracy in indoor non-line-of-sight (NLOS) environments using ultra-wideband (UWB) and angle of arrival (AoA) technologies. It examines the application of moving average filters and the adaptive offset cancellation (AOC) method in known target's location area scenario. Furthermore, this study evaluates the performance of positioning accuracy using various input membership functions in fuzzy inference systems for aera recognition in unknown target's location area situation, in conjunction with the AOC method. Experimental results show that the AOC method effectively reduces positioning errors by an average of 29.38 cm across twelve test points when the area where the target is located are known, achieving an error reduction to within 20 cm. In cases where target's location area is unknown, the fuzzy inference system using fuzzy triple std as input membership function achieves an average regional recognition accuracy of 95.68%, outperforming other methods. The proposed fuzzy inference combined with AOC (FAOC) method improves the average positioning error by 40.9% compared to the original positioning method, reducing from 69.03 cm to 44.48 cm.

## 1. Introduction

With the development of smart cities and Internet of Things (IoT) technologies, there is a high demand for indoor positioning services. Positioning refers to the process of estimating the location of mobile devices or individuals relative to their surrounding working area during motion. In outdoor environments, GPS (Global Positioning System) is a highly effective technology for locating mobile devices or personnel in activities such as driving navigation and drone positioning. GPS is a satellite-based technology that provides accurate information on the longitude, latitude, and altitude of any location on Earth. Current-ly, outdoor positioning systems have an error margin of approximately

**Data availability statement:** All relevant data are within the paper and its Supporting Information files.

**Funding:** This study is supported by National Science of Technology Council (NSTC), Taiwan, with Grant no. 112-2221-E-324-010 – and 114-2425-2221-H-028-003-.

**Competing interests:** The authors have declared that no competing interests exist.

5–10 meters [1]. However, in indoor environments, the signal weakening caused by shadowing fading and multipath effects can significantly decrease positioning accuracy, limiting the utility of GPS for indoor positioning purposes [2].

In order to provide reliable indoor positioning, various ground-based wireless communication systems have been explored for indoor positioning technologies. Common indoor positioning technologies include Wi-Fi and Bluetooth technologies widely used in any connected mobile terminals for positioning [3–7], ZigBee [8], as well as architecture modified through Radio Frequency Identification (RFID) [9] to not only provide identification but also target location information [10,11]. Although these wireless communication standards were designed for purposes other than positioning and ranging, they can still be utilized for positioning at the cost of certain performance limitations such as shorter distances, low or moderate accuracy, and limited detectability coverage. Unlike traditional positioning technologies that rely on signal strength received by target objects to meet in-door high-precision requirements, Ultra-Wideband (UWB) positioning technology com-pliant with the IEEE 802.15.4–2011 standard is a novel short-range, high-rate wireless communication technology [12–14]. It employs ultra-low duty cycle impulse pulses as the information carrier using carrier-free spread spectrum technology. Characterized by high temporal resolution, strong anti-interference capabilities, and wide bandwidth, it can achieve centimeter-level positioning accuracy [15], and is gradually being applied in the high-precision positioning field [16].

Ultra-Wideband (UWB) positioning methods commonly include Received Signal Strength Indication (RSSI), Two-Way Time-of-Flight (TW-ToF), Time of Arrival (ToA), Time Difference of Arrival (TDoA), and Angle of Arrival (AoA) [17]. These methods offer various techniques for estimating distances and determining transmitter locations. RSSI measures the power of received signals, with signal strength inversely proportional to distance [18–20]. RSSI estimates location through triangulation by analyzing the signal strength received by readers. [21,22]. However, it is sensitive to environmental factors like obstacles and interference, limiting its precision in high-accuracy applications. This technology relies on existing empirical propagation models [23], with more complex models needed for higher precision. TW-ToF calculates distance using the round-trip time of signals between a transmitter and receiver, offering higher accuracy and lower costs but facing challenges in environments with high latency or multipath effects [24]. The ToA method measures the time between transmission and reception of a signal to estimate distance. It requires precise clock synchronization but is heavily impacted by multipath effects, making it complex to implement in real-world environments [25,26]. TDoA, on the other hand, uses multiple receivers to measure the time differences in signal arrival, allowing for high-precision positioning with minimal interference. However, it demands synchronized receivers and increased hardware, adding complexity and cost [27,28]. Lastly, AoA estimates the transmitter's location by measuring the angle at which signals arrive, often using antenna arrays. While it provides high precision and performs well in multipath environments, AoA accuracy depends heavily on environmental factors like antenna layout and blockage [29,30].

Although UWB has higher positioning accuracy, UWB signals are susceptible to reflection, diffraction, and obstruction during transmission, resulting in weakened signals. Additionally, multipath interference and non-line-of-sight (NLOS) conditions can also cause significant errors in positioning results. Recent advancements in the field have further highlighted the critical need for robust NLOS mitigation strategies. Comprehensive surveys [ 31,32] indicate that while UWB remains a dominant technology for high-precision indoor localization, current research trends are shifting towards integrating adaptive algorithms to handle environmental uncertainties [33]. Notably, contemporary studies have successfully demonstrated the efficacy of hybrid approaches that incorporate fuzzy logic for NLOS identification and mitigation, proving that fuzzy-based inference remains a powerful tool for enhancing robustness in complex multipath environments [ 34,35]. This paper will investigate the performance of moving average (MA) filtering and Adaptive Offset Cancellation (AOC) to reduce positioning errors in NLOS environments using a UWB combined with AOA positioning method. However, in practical applications, the location of the tag is unknown. Therefore, we propose a fuzzy inference system (FIS) to infer the location of the tag and conduct detailed research and analysis on the positioning errors after combining with AOC.

The primary motivation of this work is to address the dual challenge in practical UWB/AoA positioning: mitigating NLOS-induced errors when the target zone is known, and accurately identifying the target zone when it is unknown. To this end, the novelty of this paper is threefold: (1) We propose the Adaptive Offset Cancellation (AOC) method to compensate for systematic positioning errors in known target zones. (2) We introduce a novel fuzzy inference system utilizing a "fuzzy triple standard deviation" input membership function, which is tailored to handle the biased and overlapping coordinate distributions in NLOS settings, for high-accuracy area recognition when the target zone is unknown. (3) We integrate these two components into a unified FAOC (Fuzzy-AOC) framework, which first identifies the target zone and then applies zone-specific error compensation, demonstrating significant overall positioning accuracy improvement in real-world-like scenarios. Moreover, the proposed FAOC framework is designed to be compatible with existing commercial UWB hardware and can be integrated into current indoor positioning systems with minimal software adaptation, facilitating its practical deployment in real-world applications.

The remaining part of this paper is outlined as follows: Section 2 discusses the related materials and methods. Section 3 focuses on the application of adaptive offset cancellation for improving positioning error. Section 4 addresses the use of fuzzy inference system for indoor area recognition. Finally, Section 5 presents the conclusions.

## 2. Materials and methods

### 2.1 Ultra-wideband and wireless channel model

Ultra-Wideband (UWB) technology is characterized by its broad frequency range, spanning from 3.1 GHz to 10.6 GHz, as defined by international standards. This extensive spectrum allows for the further division of UWB into two sub-bands: the low-frequency band (3.1 GHz to 4.8 GHz) and the high-frequency band (6.5 GHz to 10.6 GHz). The low-frequency band plays a key role in UWB applications, particularly in indoor environments [36]. Signals within this range can penetrate obstacles with greater depth, making them ideal for indoor positioning and short-range communication. Additionally, UWB signals in the low-frequency band help mitigate the impact of multipath fading, enhancing the reliability of both communication and positioning systems. On the other hand, the high-frequency band is used primarily for high-speed data transmission and radar applications. It offers higher transmission rates, enabling faster data speeds and greater bandwidth. However, signals in this band are more susceptible to interference due to lower penetration capabilities compared to the low-frequency band. Overall, UWB technology offers great potential in areas like indoor positioning, the Internet of Things (IoT), and wireless communication. Its ability to operate across a wide frequency spectrum makes it adaptable to various environments, providing benefits like resistance to interference and high time resolution. UWB technology is governed by various global standards to ensure device interoperability and consistency. A key standard is IEEE 802.15.4a [37], which defines the physical layer (PHY) and medium access control (MAC) of UWB and its compatibility with other wireless technologies. This standard outline transmission modes, modulation schemes, frequency ranges, and power limits and

specifies data formats, synchronization, and node management. IEEE 802.15.4a identifies two main UWB modulation schemes: Direct Sequence Ultra-Wideband (DS-UWB), which uses ultra-wide pulse signals for transmission and positioning, and Orthogonal Frequency Division Multiplexing (OFDM), which divides bandwidth into subcarriers for high-speed data transfer. UWB is also regulated by international organizations like the International Telecommunication Union (ITU) and the Federal Communications Commission (FCC), ensuring global coordination on frequency ranges and power limits.

The wireless channel may involve line-of-sight (LOS) propagation or be influenced by various factors, such as interference from multi-path effects, which occur when signals pass through buildings. Additionally, changes in the relative positions of transmitting and receiving points can lead to the Doppler effect, causing channel characteristics to vary over time and resulting in unstable signal quality. These phenomena, caused by either multipath or Doppler effects, are collectively referred to as fading [38]. Fading can be categorized into two main types: large-scale fading and small-scale fading. Large-scale fading is influenced by factors like distance and obstacles in the signal path. It is a slow-varying phenomenon that affects long-distance wireless communications. Techniques such as adaptive antenna gain and relay station deployment can help reduce large-scale fading effects, with shadowing often occurring over long distances, following a log-normal distribution. In contrast, small-scale fading occurs within large-scale fading areas, involving multiple signal paths over short distances. This results in rapid fluctuations in signal strength, influenced by multipath and Doppler effects, which can severely impact communication systems. Effective modeling and compensation are crucial to mitigate small-scale fading for optimal system performance. Line-of-sight (LOS) propagation occurs when radio waves travel directly between transmitting and receiving points. Non-line-of-sight (NLOS) propagation, where obstacles obstruct the direct path, can significantly affect signal quality, especially in indoor positioning systems [39].

## 2.2 Angle of arrival positioning algorithm

Most positioning algorithms rely on at least three base stations to establish a positioning system, leading to a more complex and costly system setup. In the study documented in [40], a positioning method based on the Angle of Arrival (AoA) algorithm using a single base station is proposed as shown in Fig 1. This method significantly reduces the setup cost by requiring only a single base station.

The AoA positioning technique primarily involves measuring the angle of arrival between the target tag and the base station. Utilizing a directional antenna originating from the base station, which must pass through the tag, the intersection point of the two rays indicates the tag's location. A drawback of this algorithm is the increased error in NLOS environments. When the base station is equipped with an antenna array, the array determines the angle of incidence based on the signal transmitted by the target object. The angles of incidence for two antennas are denoted as $\alpha_1$ and $\alpha_2$. By using these two angles with the antennas as reference points, the intersection of the lines constructed in the direction of incidence represents the location of the target object. Assuming the coordinates of the target object are $(T_x, T_y)$ and the coordinates of the antennas are $(x_i, y_i)$, with two antennas labeled as $(x_1, y_1)$ and $(x_2, y_2)$, their positions satisfy the geometric properties of AoA shown below,

$$\mathbf{AW} = \mathbf{Z},\tag{1}$$

where $\mathbf{A} = \begin{bmatrix} 1 & -\tan\alpha_1 \\ 1 & -\tan\alpha_2 \end{bmatrix}$, $\mathbf{W} = \begin{bmatrix} T_x \\ T_y \end{bmatrix}$, $\mathbf{Z} = \begin{bmatrix} y_1 - x_1 \cdot \tan\alpha_1 \\ y_2 - x_2 \cdot \tan\alpha_2 \end{bmatrix}$. After performing matrix operations on Equation (1), we could obtain the solution by

$$\mathbf{W} = \mathbf{A}^{-1}\mathbf{Z}.\tag{2}$$

## 2.3 Adaptive offset cancellation

During the experimental process of measuring errors, we observed that the positioning errors in each measurement varied due to environmental changes. Therefore, by conducting measurements over a period of time and collecting sufficient

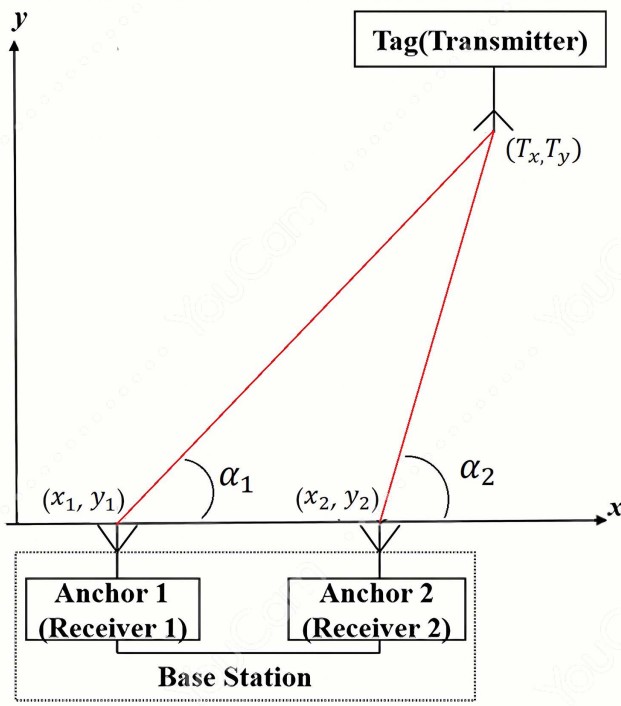

**Fig 1. Illustration of AoA positioning method.**

data, we identified the average offset of the x and y coordinates of test points in different areas. Then, during testing, we compensated for the coordinate measurement values of the test point by the average offset of its corresponding area to reduce positioning errors. We named this method adaptive offset cancellation (AOC) [41,42].

The calculation method for the average offset of the *x* and *y* coordinates of the *i*-th training test point can be expressed by

$$X_{O,i} = \bar{x}_{train,i} - T_{x,i}, \tag{3}$$

and

$$Y_{O,i} = \bar{y}_{train,i} - T_{y,i}, \tag{4}$$

respectively, where $\bar{x}_{train,i}$ and $\bar{y}_{train,i}$ represent the average coordinates after applying moving average filtering to the training data, and $T_{x,i}$ and $T_{y,i}$ are the true coordinates of the test point for x and y, respectively.

After obtaining the offsets of x and y in each region, the experiment first processes the measured signals using moving average filtering, and then corrects them based on the offset of the region where the test point is located. The corrected coordinates of the test point $(\acute{x}_i, \acute{y}_i)$ are

$$\acute{x}_i = \bar{x}_i - X_{O,i}, \tag{5}$$

$$\acute{y}_i = \bar{y}_i - Y_{O,i}, \tag{6}$$

where $X_{O,i}$ and $Y_{O,i}$ are the offsets of the x and y coordinates for each test point, where $i = 1 \sim 12$, and $\bar{x}_i$ and $\bar{y}_i$ are the average coordinates after applying moving average filtering to the $i$-th test point.

## 2.4 Fuzzy inference system

Fuzzy inference originated in the 1960s when Professor L.A. Zadeh introduced fuzzy logic to address real-world phenomena that involve uncertainty and imprecision, which traditional Boolean logic struggles to manage. Unlike Boolean logic, fuzzy logic allows conclusions to reflect varying degrees of fuzziness and uncertainty, making it applicable to a wide range of practical problems [43–45]. A fuzzy inference system, as shown in Fig 2, typically consists of five main components: Fuzzification, Fuzzy Inference Engine, Database, Fuzzy Rule Base, and Defuzzification.

In the fuzzification stage, the system's primary task is to define fuzzy sets and map input variables to these sets using membership functions. These membership functions, which are stored in the Database, are commonly represented by triangular, trapezoidal, or Gaussian shapes [42]. In this study, trapezoidal membership functions are used for input, while triangular membership functions are employed for output.

During the Fuzzy Inference Engine stage, fuzzy input sets are processed and mapped to fuzzy output sets using membership functions and a fuzzy rule base. This rule base contains "if-then" rules that define the relationship between fuzzy conditions and their corresponding outcomes. Finally, in the defuzzification stage, the system converts the fuzzy results into precise numerical values or operations for further evaluation and decision-making [46]. Various defuzzification methods can be applied, with the Centroid Method being one of the most widely used.

The Centroid Method calculates the defuzzification result based on the center or centroid of the fuzzy set, positioning the result at the center point of the fuzzy set according to its distribution. The other defuzzification method is the Max-Membership Value Averaging approach. In this method, the maximum membership value for each fuzzy rule's output is first determined. Then, the highest of these maximum values is selected, and the average of these values is calculated to obtain the final fuzzy output.

## 2.5 Experimental hardware setup

The UWB ranging and communication were implemented using Decawave DW1000 transceiver modules (compliant with IEEE 802.15.4–2011 UWB standard). A custom-designed four-element linear antenna array with an element spacing of 5 cm was used at the base station to enable AoA estimation. The tag node consisted of a single DW1000 module with an omnidirectional antenna. An ESP32 microcontroller served as the processing unit on both the base station and the tag, handling signal timing, data packaging, and communication. The base station communicated with a central processing

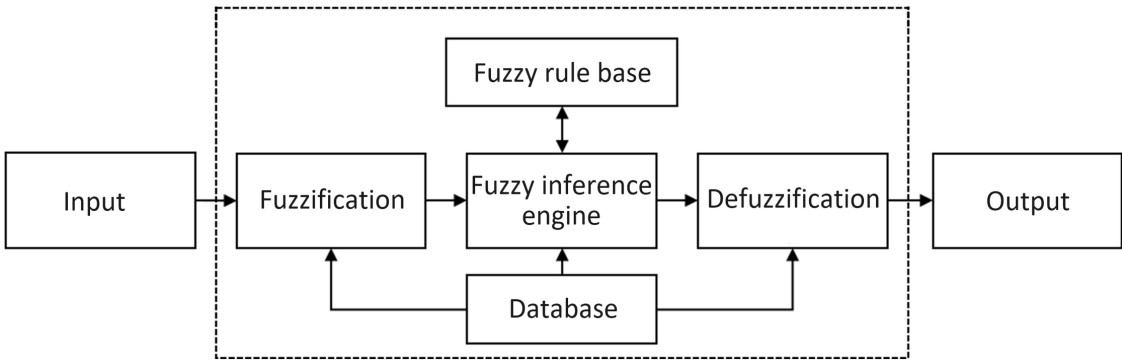

**Fig 2. Architecture of fuzzy inference system.**

server via a Wi-Fi module (ESP32 integrated) using WebSocket protocol for real-time data transfer. All devices were mounted on tripods at a fixed height of 1.2 meters to maintain consistency throughout the experiments. It is worth noting that the proposed FAOC method is implemented at the software level and is fully compatible with commercially available UWB systems, such as those based on Decawave DW1000 modules. The fuzzy inference engine can be executed on low-cost microcontrollers (e.g., ESP32), making it suitable for embedded deployment without requiring hardware modifications.

## 3 Adaptive offset cancellation for indoor positioning

During experiments, it was found that the positioning errors of the test points vary due to daily environmental changes. Therefore, we propose the AOC method with the intention of collecting sufficient data to determine the offsets of the x and y coordinates within each area. During testing, compensating for the measured offsets is applied to reduce positioning errors.

### 3.1 Experimental environment

The experimental environment in this study is a space measuring 600 cm by 800 cm. We divided this space into 12 uniformly distributed areas and placed a test point (TP) at the center of each area, as shown in Fig 3. This uniform distribution of test points across the experimental area allows for a comprehensive evaluation of the positioning system. Points at varying distances and angles relative to the base station are included. Specifically, test points located at the far corners of the environment (e.g., **TP$_C$**, **TP$_F$**, **TP$_I$**, **TP$_L$**) represent more demanding NLOS scenarios due to extended signal travel distances and increased potential for signal blockage and multipath interference.

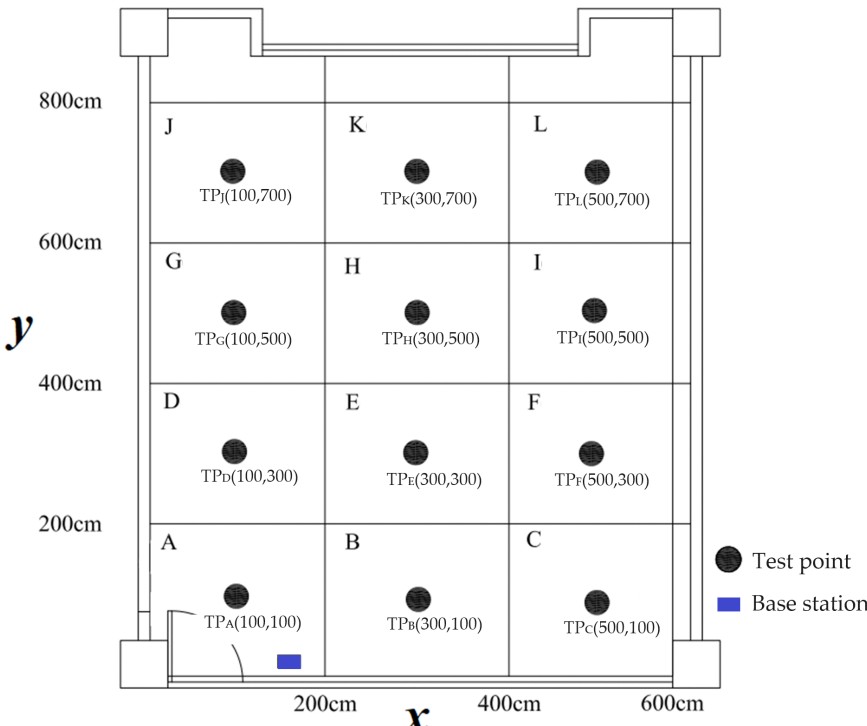

**Fig 3. Illustration of the experimental environment.**

The coordinates of the base station (BS) are (150,14). The coordinates and distance data between the BS and the TPs in each area are shown in Table 1. A positive angle in Table 1 indicates that the TP is to the right of the two antennas at the BS, while a negative angle suggests that the TP is to the left at the BS.

### 3.2 Experimental procedures

The experimental procedure is intentionally structured into two distinct phases—training and testing—to emulate a practical deployment paradigm: an initial long-term calibration period followed by operational use. This separation ensures that the system's error compensation is learned from comprehensive calibration data, preventing overfitting and providing a realistic assessment of performance in a scenario akin to real-world application.

- Training Phase: The training phase aimed to establish a robust, zone-specific error model. Data collection spanned seven days to capture a wide range of environmental variations. At each of the 12 test points, $TP_A$ to $TP_L$, 7,000 consecutive positioning samples were collected at a sampling rate of 1 Hz. This extensive dataset ensures statistical significance for subsequent analysis. The procedure began with the UWB tag transmitting signals to the base station (BS). The BS, equipped with an antenna array, calculated the raw tag coordinates, angles, and distance information using the AoA algorithm. This raw data stream was transmitted to a central server via Wi-Fi. A moving average filter with a window size of 10, as optimized in Section 3.3 and Fig 4, was then applied to the raw coordinates of this long-term data to suppress high-frequency noise and yield a stable, representative average position ($\bar{x}_{train,i}$, $\bar{y}_{train,i}$) for each point. The systematic error, or zone-specific offset ($X_{O,i}, Y_{O,i}$), was computed as the difference between this filtered average position and the known true coordinates ($T_{x,i}, T_{y,i}$) of the training test point, seeing Eqs. 3 and 4. These 12 offset pairs, each uniquely associated with a zone label, A through L, were stored in a lookup table. The proposed training procedure is shown in Fig 4.

- Testing Phase: The testing phase evaluated the system's positioning accuracy using a separate dataset. A comparable volume of test samples, 7,000 samples per point at 1 Hz, was collected to match the training data's statistical scale. In this phase, the tag transmits signals in real-time. The BS calculates instantaneous coordinates using AoA, which are immediately processed by the same moving average filter with window size 10. The primary role of the filter here is to provide a smoothed, real-time coordinate estimate by mitigating instantaneous measurement noise, thus preparing a cleaner signal for the subsequent correction step. The system then proceeds under one of two conditions: (i) for

Table 1. The coordinates, distance and angle data between the TPs and the BS.

| Test point | Coordinates(cm) | Distance from the BS (cm) | Angle to the BS antenna. (°) |
|---|---|---|---|
| $TP_A$ | (100,100) | 93.3 | −31.8° |
| $TP_B$ | (300,100) | 170.3 | 61.9° |
| $TP_C$ | (500,100) | 359.99 | 77.24° |
| $TP_D$ | (100,300) | 283.75 | −9.8° |
| $TP_E$ | (300,300) | 317.73 | 28.4° |
| $TP_F$ | (500,300) | 448.77 | 51.5° |
| $TP_G$ | (100,500) | 481.99 | −5.82° |
| $TP_H$ | (300,500) | 502.74 | 17.49° |
| $TP_I$ | (500,500) | 594.3 | 36.2° |
| $TP_J$ | (100,700) | 681.26 | −4.2° |
| $TP_K$ | (300,700) | 696.1 | 12.54° |
| $TP_L$ | (500,700) | 764.84 | 27.33° |

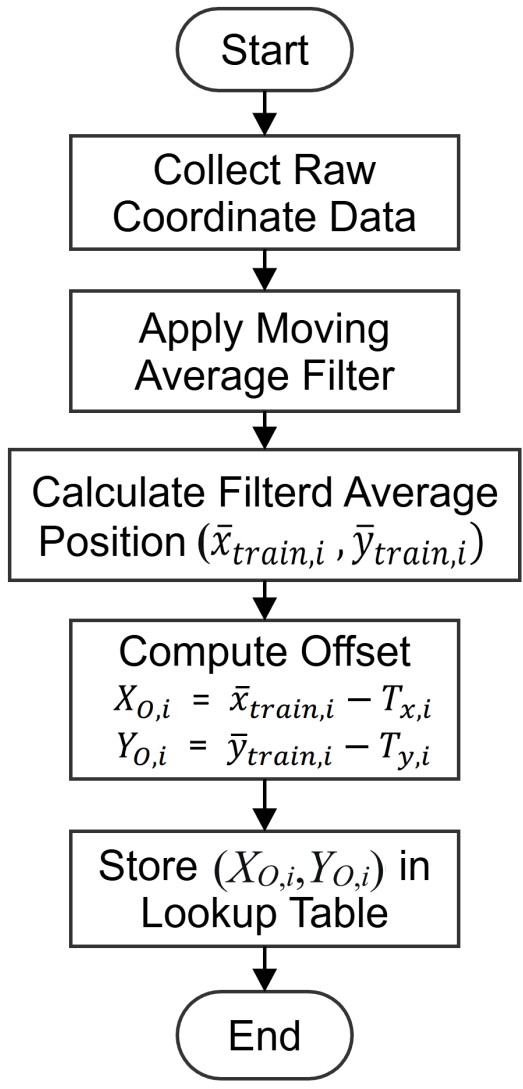

**Fig 4. Proposed training procedure for parameter estimation.**

evaluating the AOC method alone, the true zone of the tag is assumed to be known, and the corresponding pre-stored offset is retrieved from the lookup table; (ii) For evaluating the integrated FAOC method, the smoothed coordinates are first input to the Fuzzy Inference System detailed in Section 4 to estimate the tag's zone. Finally, the offset pair corresponding to the identified zone is fetched and subtracted from the smoothed coordinates, as specified by Equations 5 and 6, producing the final corrected position.

### 3.3 Experiment on moving average filtering

The moving average filter is a common signal processing technique used to remove high-frequency noise from the signal, smooth the signal, and improve the signal-to-noise ratio. Tests were performed at test points A, C, J, and L, located in the four corner areas of the experimental environment, to examine the variations in RMSE across different window sizes, as shown in Fig 5.

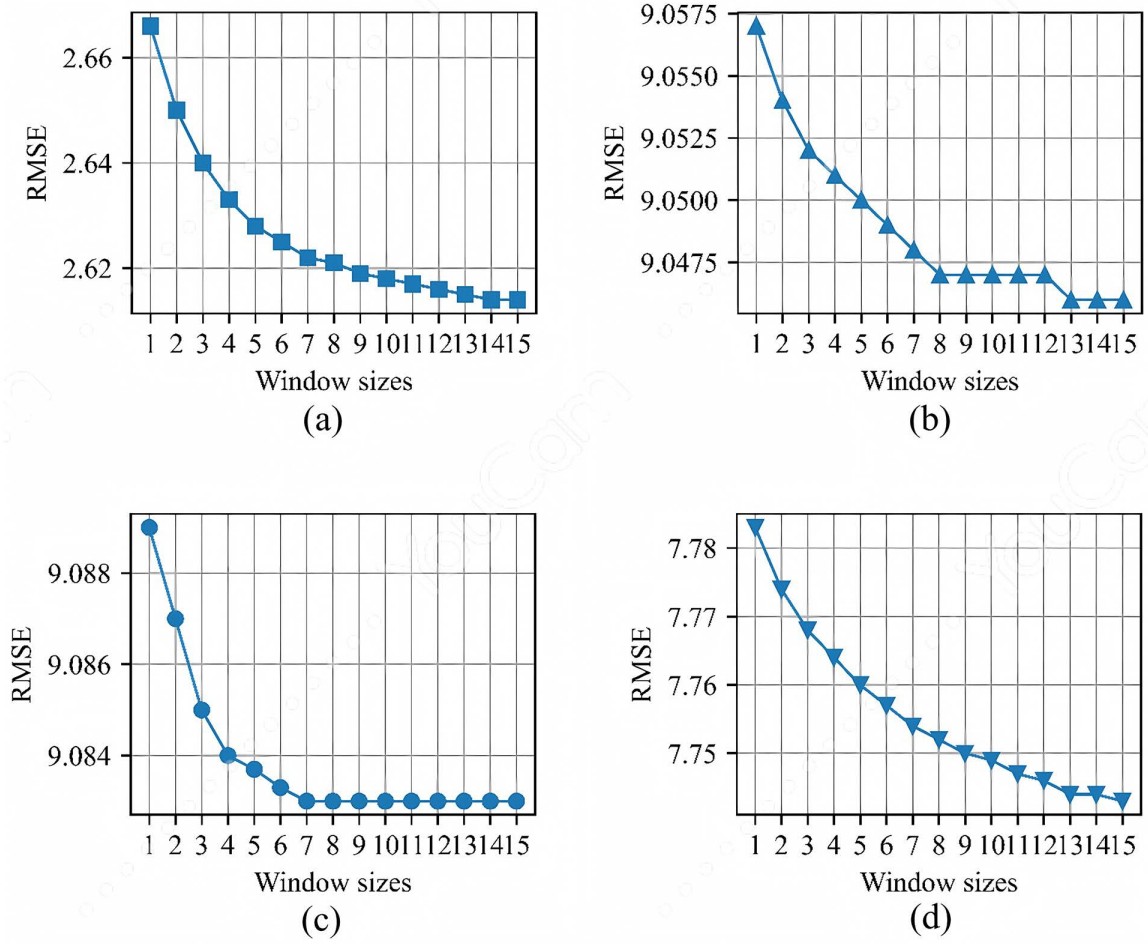

**Fig 5. Variations in RMSE for different window sizes at TPs(a) A, (b) C, (c) J, and (d)L.**

### 3.4 Experiment on adaptive offset cancellation

In this section, we apply adaptive offset elimination in an NLOS environment to analyze the experimental results. The offset values obtained during the training phase for each test point are then used with the adaptive offset cancellation technique to remove these values for the corresponding test points. Fig 6 shows the cumulative distribution function (CDF), which illustrates the positional error distribution at test points A, C, J, and L, located in the four corners of the experimental setup, evaluated using four different methods.

In Table 2, it can be observed that the positioning errors of the three test points, $TP_A$, $TP_D$, and $TP_G$ in front of the base station are lower than those of the other test points. This is because some are farther from the base station, while others exceed the effective angle range of AoA. Although $TP_J$ is located in front of the base station, its positioning error is larger due to being in NLOS environment and being farther from the base station. However, the adaptive offset elimination method can reduce its positioning error by approximately 20 centimeters. Besides, the test points located on the edge of the experimental area and farther from the base station, such as $TP_C$, $TP_I$, and $TP_L$, have the highest positioning errors, reaching potentially close to 100 centimeters. Nevertheless, after applying the adaptive offset elimination method, it can

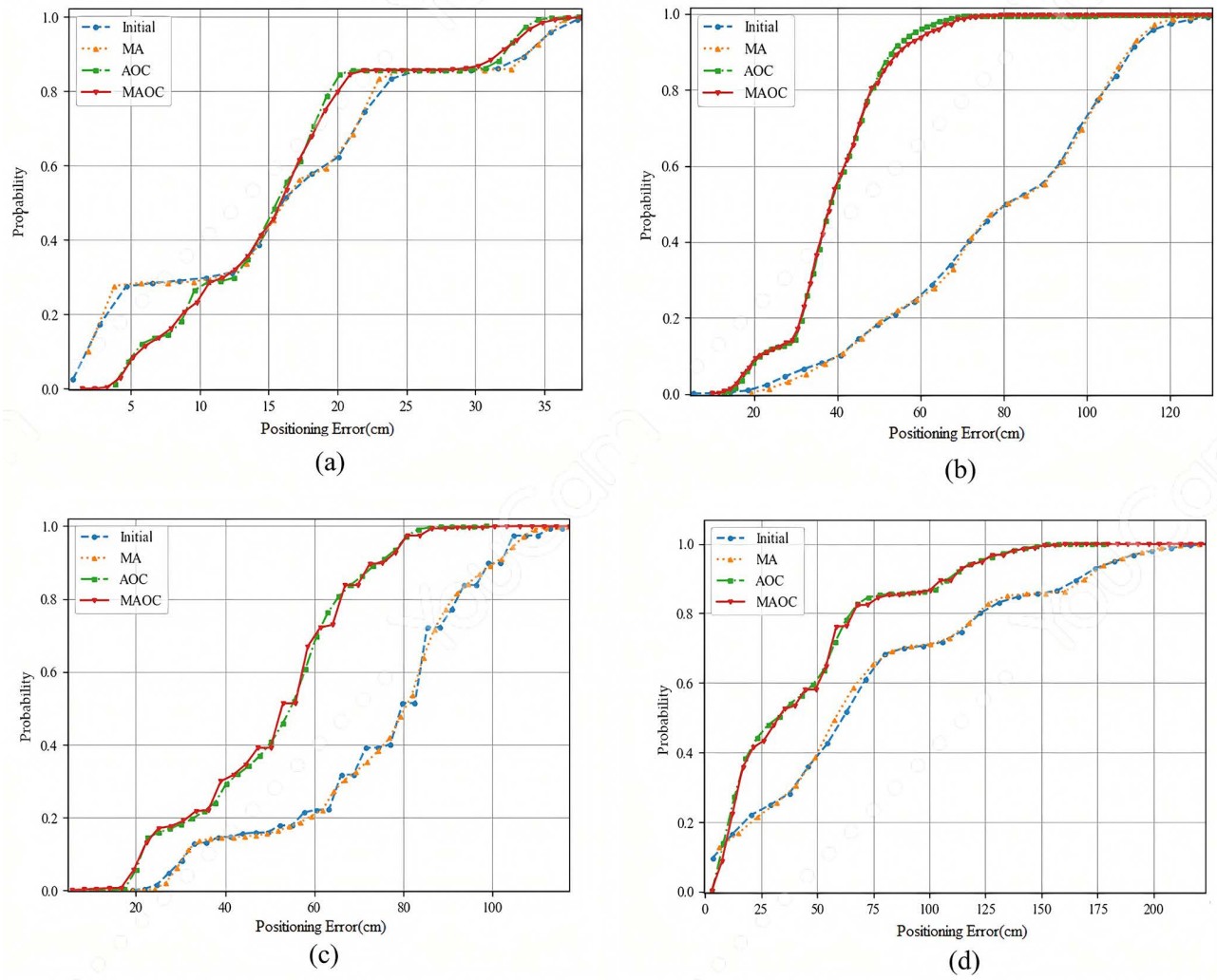

**Fig 6. The CDFs of different positioning methods in NLOS at TPs(a) A, (b) C, (c) J, and (d)L.**

be observed that the positioning errors for these three test points, $TP_C$, $TP_I$, and $TP_L$, can be reduced by approximately 30–50 centimeters.

The results in Table 2 demonstrate that when the locations of target areas are known, the proposed AOC method effectively mitigates positioning errors induced by NLOS environments, resulting in an average reduction of 29.38 cm in positioning error across twelve test points, achieving the high precision with the position error below 40 cm.

### 3.5 Discussion on AOC robustness in diverse NLOS environments

The AOC method demonstrates consistent error reduction across various NLOS conditions encountered in our experimental setup, which included walls, office partitions, and intermittent human obstructions. As shown in Table 2, test points located behind obstacles (e.g., $TP_C$, $TP_F$, $TP_I$) still achieved significant error reduction, confirming the method's applicability in common indoor NLOS scenarios.

**Table 2. The average positioning errors of TPs in the NLOS environment using different methods.**

| $TP_i$ | $\overline{e}_{l,i}$ | $\overline{e}_{MA,i}$ | $\overline{e}_{AOC,i}$ | $\overline{e}_{MAOC,i}$ |
|---|---|---|---|---|
| $TP_A$ | 17.87 | 17.8 | 17.09 | 17.04 |
| $TP_B$ | 73.93 | 73.88 | 39.42 | 39.34 |
| $TP_C$ | 83.81 | 83.65 | 41.67 | 41.12 |
| $TP_D$ | 18.27 | 18.15 | 15.97 | 15.82 |
| $TP_E$ | 90.64 | 90.63 | 61.46 | 61.37 |
| $TP_F$ | 97.1 | 97.08 | 35.75 | 35.52 |
| $TP_G$ | 30.73 | 30.6 | 11.64 | 11.2 |
| $TP_H$ | 62.53 | 62.46 | 30.98 | 30.64 |
| $TP_I$ | 113.33 | 113.26 | 66.12 | 66 |
| $TP_J$ | 76.73 | 76.72 | 53.37 | 53.32 |
| $TP_K$ | 83.44 | 83.36 | 57.51 | 57.21 |
| $TP_L$ | 81 | 80.82 | 48.85 | 48.28 |
| Average | 69.12 | 69.03 | 39.99 | 39.74 |

However, the performance of AOC is influenced by the spatial consistency of systematic errors within each zone. In environments where error patterns change rapidly due to highly dynamic or metallic-dense obstacles, the pre-calibrated offset values may require more frequent updates. Future implementations could integrate real-time environmental sensing or adaptive learning mechanisms to maintain accuracy in such challenging conditions.

Nevertheless, for most typical indoor environments—such as offices, warehouses, and smart homes—the AOC method provides a computationally efficient and effective means of mitigating NLOS-induced positioning errors.

## 4 Fuzzy inference system for indoor area recognition

The experimental results from Section 3 demonstrate that the positioning errors in indoor environments can be reduced using the adaptive offset elimination method. However, this method requires information about the area where the target is located (target area) to eliminate offset errors. Therefore, in this section, a fuzzy inference system will first discriminate the target area and compare its accuracy with clear area recognition and *K*-means area recognition methods.

The clear area recognition method employs the boundaries of each area shown in Fig 3 as the criteria for region recognition. Furthermore, as this study divides the experimental environment into twelve zones, the K value for *K*-means method in this research will be set to 12.

### 4.1 Experimental environment and procedures

The experimental environment is an indoor space measuring 600 cm by 800 cm, divided into 12 zones, with a test point located at the center of each zone, as shown in Fig 3. The experimental framework consists of training and testing phases, corresponding to Figs 4 and 7, respectively. The training phase follows the data collection and parameter estimation procedures detailed in Section 3.2 and illustrated in Fig 4. The testing phase shown in Fig 7 shares the same initial three steps of signal acquisition and processing as the training phase. However, in the final step, the smoothed coordinates are input into the Fuzzy Inference System (FIS)—developed during the training phase—to execute region classification and determine the final target zone.

To identify a membership function with superior performance, this study uses raw x- and y-coordinate data measured in the experimental environment as inputs to the fuzzy inference system. We first derived the probability density function (PDF) of the data to understand its distribution, which served as the basis for constructing the input membership functions. Based on the experimental layout, the x-coordinate was divided into three linguistic levels: Near (N), Middle (M), and Far

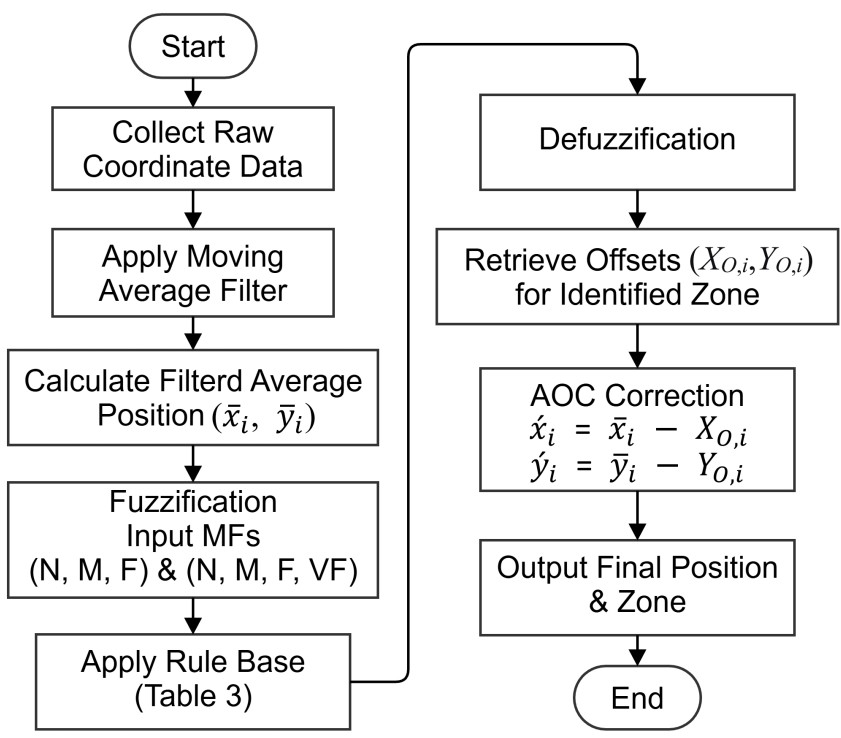

**Fig 7. Testing procedure for fuzzy area recognition.**

(F). The y-coordinate was divided into four levels: Near (N), Middle (M), Far (F), and Very Far (VF). The resulting PDFs for the x- and y-coordinate levels are shown in Figs. 8 and 9, respectively.

Building on our prior research [47,48], we compared the area recognition accuracy of three different types of membership functions—Type I, Type II, and Type III—which correspond to functions based on 2, 3, and 4 standard deviations,

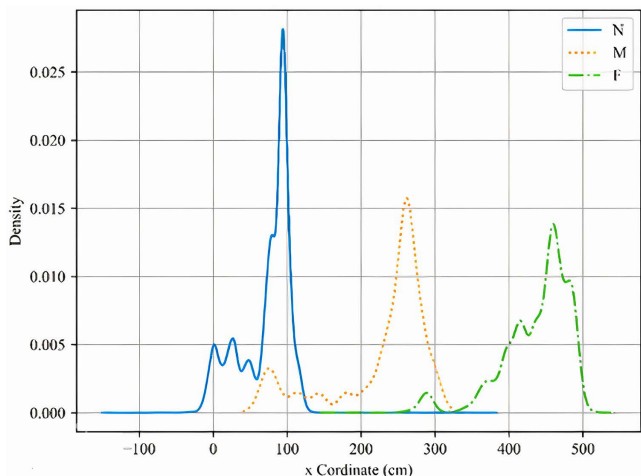

**Fig 8. The PDFs of N, M and F levels in x-coordinate.**

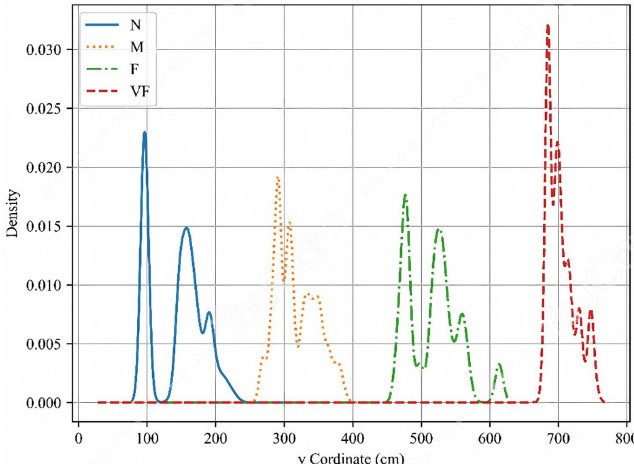

**Fig 9. The PDFs of N, M, F and VF levels in y-coordinate.**

respectively. The results, summarized in Table 3, indicate that Type II (3 standard deviations) achieved the highest recognition accuracy. This is because Type I covers a narrower range of the training data, while Type II encompasses a broader and more representative portion. Although Type III has the widest interval and covers the most data, it also introduces irrelevant data points, leading to a decrease in accuracy [48].

Based on the above analysis, we propose using an input membership function with a triple standard deviation interval (Fuzzy triple std), as illustrated in Figs 10 and 11. This configuration demonstrated the highest average area recognition accuracy and was therefore adopted for subsequent experiments in combination with the Adaptive Offset Cancellation (AOC) method. The selection of the specific "fuzzy triple standard deviation" (triple std) membership function was driven by the need to effectively model this observed data characteristic—namely, the biased and overlapping distributions caused by systematic NLOS errors. Compared to conventional symmetric membership functions (e.g., standard triangular or Gaussian), the triple std function offers two key advantages for our application: (1) It is inherently data-driven, anchored on the median and scaled by the standard deviation of the training data, which automatically accounts for the distribution's shift and spread. (2) Its asymmetric design, with core ($\pm 1.5\sigma$) and support ($\pm 3\sigma$) intervals, provides a more flexible and representative fuzzy set for data points that do not conform to idealized, symmetric clusters. This flexibility is crucial for accurately classifying coordinates in the presence of NLOS-induced uncertainty.

## 4.2 Proposed fuzzy inference system area recognition

The proposed system for indoor area recognition is a Mamdani-type fuzzy inference system (FIS) with a two-input, one-output architecture, designed for efficient and interpretable classification of measured coordinates into one of twelve predefined zones (A–L).

**Table 3. The average recognition accuracy of three different input membership functions.**

| Membership function | Average recognition accuracy |
|---|---|
| Type I (2 std) | 90.06% |
| Type II (3 std) | 95.68% |
| Type III (4 std) | 91.65% |

*Note*: Data sourced from [48]

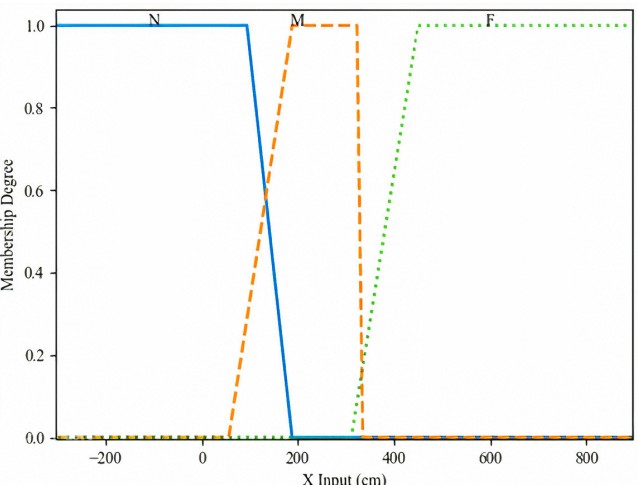

**Fig 10. The input membership functions of x.**

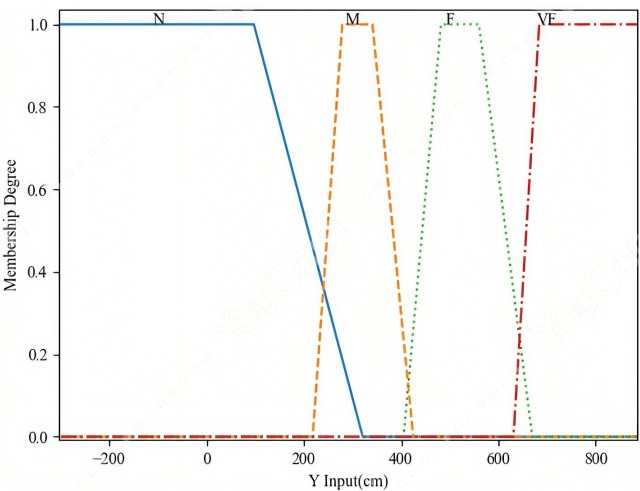

**Fig 11. The input membership functions of y.**

**4.2.1. Input stage.** In this study, the fuzzy inference system (FIS) employs two input variables: $x$ and $y$, representing the measured $x$-coordinate and $y$-coordinate of the target tag, respectively. Specifically, the $x$-coordinate is partitioned into three linguistic levels: Near (N), Middle (M), and Far (F), corresponding to the fuzzy set $F_1^{I_1} \in \{N, M, F\}$, where $I_1 = 1, 2, 3$. Similarly, the $y$-coordinate is divided into four linguistic levels: Near (N), Middle (M), Far (F), and Very Far (VF), represented by the fuzzy set $F_2^{I_2} \in \{N, M, F, VF\}$, where $I_2 = 1, 2, .., 4$, as illustrated in Figs 10 and 11.

**4.2.2. Output stage.** The fuzzy inference system (FIS) in this study has a single output variable, $z$, which represents the recognized region. This variable corresponds to a total of 12 possible regions, with its linguistic terms denoted as $F_3^{I_3} \in \{A, B, \ldots, L\}$, where $I_3 = 1, 2, \ldots, 12$. Triangular membership functions are used to define these output terms. These functions are preferred for their structural simplicity and parameter efficiency—requiring only the specification of the

peak position and the two side slopes to fully describe their shape. The resulting output membership functions are shown in Fig 12.

**4.2.3. Fuzzy rule base.** The fuzzy inference engine operates on a set of linguistic IF-THEN rules. To formalize the inference logic, the $k$-th fuzzy rule $R_k$ is represented as follows:

$$R_k : \text{IF } x \text{ is } F_1^{l_1} \text{ AND } y \text{ is } F_2^{l_2} \text{ THEN } z = F_3^{l_3}, \tag{7}$$

where $F_1^{l_1}$, $F_2^{l_2}$ and $F_3^{l_3}$ are the linguistic terms of the input variables $x$, $y$ and the output variable $z$, respectively, and $l_1 = 1, 2, 3, l_2 = 1,2, .., 4, l_3$ and the index of rule $k = 1, 2, …, 12$. According to the above rule, the corresponding region output based on the input $x$ and $y$ values is shown in Table 4.

**4.2.4. Defuzzification.** There are several defuzzification methods available to convert fuzzy inference results into a crisp output. In this study, the centroid calculation method, also known as the Center of Area (COA), is employed. This method calculates the center of gravity of the aggregated membership functions' curve to determine the most representative numerical value for the identified region. The centroid output $z_{COA}$ is determined by substituting the discrete parameters of the inference results expressed by

$$z_{COA} = \frac{\sum_{i=1}^{q} z_i \cdot \mu_{F_3^l}(z_i)}{\sum_{i=1}^{q} \mu_{F_3^l}(z_i)}, \tag{8}$$

where $q$ denotes the total number of predefined zones ($q = 12$ in this study), $z_i$ represents the discrete values in the fuzzy set of inference results—corresponding to the numerical identifiers of the 12 predefined zones—and $\mu_{F_3^l}(z_i)$ denotes the degree of membership for the $i$-th discrete value. Through this process, the fuzzy linguistic labels derived from the rules in Table 4 are transformed into a singular identified zone label for subsequent error compensation.

## 4.3 Fuzzy inference system combined with adaptive offset cancellation

The experimental results in section 3 demonstrate that the AOC method can improve the positioning errors of test points, but such improvement can only be carried out once the true target area is known. However, in practical applications, the

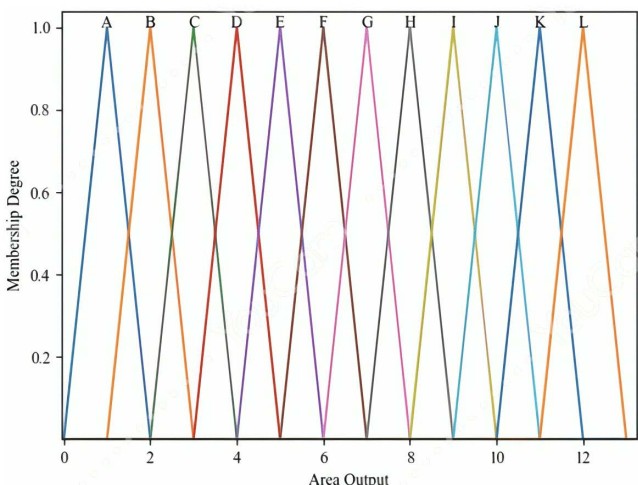

**Fig 12. The output membership functions.**

**Table 4. The output of the fuzzy rule used for FIS area recognition.**

| yx | N | M | F | VF |
|---|---|---|---|---|
| N | A | D | G | J |
| M | B | E | H | K |
| F | C | F | I | L |

target area is generally not known in advance. This section will first identify the location of the target area using the Fuzzy triple std method, and then conduct experiments using the AOC method. The experimental results are shown in Table 5, including the region recognition accuracy of the FIS using Fuzzy triple std in 12 areas, the original positioning error (Original), the positioning error after improvement by the AOC method (AOC), and the positioning error using the Fuzzy triple std region recognition method combined with the AOC method (FAOC).

In Table 5, the original positioning error (Original) represents the unprocessed positioning error of the test data. The positioning error after using the AOC method (AOC) is the result of the experiment in Section 3, assuming the known location of the target area and applying the AOC method for positioning error improvement. The improved positioning error by the Fuzzy triple std combined with the AOC method (FAOC) involves using Fuzzy triple std as the input membership function of the FIS for regional recognition of the target area when the location is unknown, and then applying the AOC method based on the recognition results.

As shown in Table 5, Areas A, B, D, and H achieve a 100% recognition rate, resulting in identical positioning accuracy for both AOC and the proposed FAOC methods in these regions. In contrast, for the remaining regions where recognition rates exceed 90% but fall short of 100%, the positioning error of FAOC is slightly higher than that of AOC. This discrepancy arises from area misclassifications; even with high recognition rates, an incorrect regional judgment introduces additional error. Consequently, the performance gap between FAOC and AOC widens as the fuzzy recognition rate decreases. Overall, the proposed FAOC method—which sequentially identifies the target area via the fuzzy inference system before applying AOC—successfully reduces the average positioning error across all twelve regions from 69.03 cm to 44.48 cm, achieving a significant improvement rate of 40.9%.

**Table 5. The accuracy of fuzzy area recognition in different regions and positioning error using different methods. (unit: cm).**

| $TP_i$ | Fuzzy accuracy | Positioning error (cm) | | |
|---|---|---|---|---|
| | | Original | AOC | FAOC |
| $TP_A$ | 100% | 17.8 | 17.04 | 17.04 |
| $TP_B$ | 100% | 73.88 | 39.34 | 39.34 |
| $TP_C$ | 99.7% | 83.65 | 41.12 | 41.82 |
| $TP_D$ | 100% | 18.15 | 15.82 | 15.82 |
| $TP_E$ | 85% | 90.63 | 61.37 | 73.05 |
| $TP_F$ | 89.8% | 97.08 | 35.52 | 48.8 |
| $TP_G$ | 99.8% | 30.6 | 11.2 | 11.31 |
| $TP_H$ | 100% | 62.46 | 30.64 | 30.64 |
| $TP_I$ | 90.1% | 113.26 | 66 | 73.02 |
| $TP_J$ | 99.9% | 76.72 | 53.32 | 53.33 |
| $TP_K$ | 91% | 83.36 | 57.21 | 63.87 |
| $TP_L$ | 92.9% | 80.82 | 48.28 | 65.74 |
| Average | 95.68% | 69.03 | 39.74 | 44.48 |

## 4.4 Scalability and applicability in complex environments

While the experiments in this study were conducted in a controlled indoor environment measuring 600 cm × 800 cm, the proposed FAOC method is designed with scalability in mind. In larger or more complex environments, such as multi-story buildings or smart city infrastructures, the system can be extended by increasing the number of zones and corresponding fuzzy rules. The modular structure of the fuzzy inference system allows for easy reconfiguration by adjusting membership functions and rule bases. For multi-story scenarios, a hierarchical fuzzy system could be implemented to first identify the floor level and then the specific zone within that floor. Furthermore, the computational efficiency of the FAOC method makes it suitable for distributed deployment across multiple base stations, enabling seamless coverage expansion. These scalability features position the FAOC method as a viable solution for a wide range of indoor positioning applications.

## 5. Conclusions

This study presents a comprehensive framework for enhancing UWB/AoA-based indoor positioning accuracy in NLOS environments by addressing both area recognition and error compensation. Experimental results demonstrate that the proposed AOC method effectively mitigates NLOS-induced errors, reducing the average positioning error by 29.38 cm and achieving sub-40 cm precision when the target area is known. In scenarios with unknown locations, the fuzzy inference system using 'Fuzzy triple std' membership functions achieved a 95.68% recognition accuracy, outperforming both the direct boundary-based 'Clear' method (85.21%) and the K-means approach (95.45%). Overall, the integrated FAOC method reduced the average positioning error from 69.03 cm to 44.48 cm, representing a 40.9% improvement.

Building on these promising results, our future research will first address the challenges of real-time environmental dynamics and long-term stability. We recognize that the current AOC method, which relies on pre-calibrated offsets, may have limitations in highly dynamic scenarios. Therefore, we plan to develop an online adaptive offset update mechanism. This mechanism will continuously monitor incoming positioning data and dynamically refine zone-specific offset values using sliding window estimation or recursive learning techniques. By doing so, the system will be capable of adapting to instantaneous environmental changes—such as moving obstacles or varying multipath conditions—while maintaining accuracy over extended periods without requiring manual recalibration. This advancement will transform the current FAOC framework into a truly adaptive real-time positioning system.

In parallel, we will pursue several avenues to further enhance positioning accuracy towards sub-20 cm levels across all zones. First, we plan to integrate fuzzy logic with advanced machine learning techniques, such as deep neural networks and hybrid neuro-fuzzy systems (e.g., ANFIS), to improve NLOS identification and mitigation. Comparative studies with state-of-the-art machine learning and big data-based localization models will also be conducted to validate and refine our approach. Second, from a hardware and signal processing perspective, we will explore the integration of higher-precision UWB modules, improved antenna arrays for more accurate AoA estimation, and advanced filtering techniques such as adaptive Kalman filtering. Finally, multi-sensor fusion approaches that combine UWB with inertial measurement units (IMU) or visual sensors will be investigated to complement UWB measurements in particularly challenging NLOS conditions.

Finally, to facilitate practical deployment, we will focus on scaling the FAOC framework to larger and more complex environments, such as multi-story buildings, through hierarchical fuzzy systems and distributed base station architectures. Concurrently, we aim to deploy the system on low-power embedded platforms (e.g., ARM Cortex-M series) to validate its compatibility with commercial UWB hardware and its suitability for real-world applications like smart warehouses and asset tracking.

## Supporting information

**S1 File. The data set obtained in positioning experiments.**
(XLSX)

## Acknowledgments

The authors are grateful to their home institutions for administrative support.

## Author contributions

**Conceptualization:** Shu-Hung Lee, Shu-Wai Chang, Yung-Fa Huang.

**Data curation:** Shu-Wai Chang.

**Formal analysis:** Shu-Hung Lee, Shu-Wai Chang, Yung-Fa Huang, Yung-Hoh Sheu.

**Funding acquisition:** Yung-Fa Huang, Yung-Hoh Sheu.

**Investigation:** Shu-Hung Lee, Shu-Wai Chang, Yung-Fa Huang.

**Methodology:** Shu-Hung Lee, Yung-Fa Huang.

**Project administration:** Yung-Fa Huang, Yung-Hoh Sheu.

**Resources:** Yung-Hoh Sheu.

**Software:** Shu-Wai Chang.

**Supervision:** Yung-Fa Huang, Yung-Hoh Sheu.

**Validation:** Shu-Hung Lee, Shu-Wai Chang, Yung-Fa Huang, Yung-Hoh Sheu.

**Visualization:** Yung-Hoh Sheu.

**Writing – original draft:** Shu-Hung Lee, Shu-Wai Chang, Yung-Fa Huang.

**Writing – review & editing:** Shu-Hung Lee, Yung-Fa Huang, Yung-Hoh Sheu.

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
