## [Decision Letter · Decision Letter 0]

18 Dec 2025

PONE-D-25-27749Precision improvement for indoor positioning based on fuzzy inference system with ultra-wideband wireless communicationsPLOS One

Dear Dr. Huang,

Thank you for submitting your manuscript to PLOS ONE. After careful consideration, we feel that it has merit but does not fully meet PLOS ONE’s publication criteria as it currently stands. Therefore, we invite you to submit a revised version of the manuscript that addresses the points raised during the review process.

A letter that responds to each point raised by the academic editor and reviewer(s). You should upload this letter as a separate file labeled ’Response to Reviewers’.A marked-up copy of your manuscript that highlights changes made to the original version. You should upload this as a separate file labeled ’Revised Manuscript with Track Changes’.An unmarked version of your revised paper without tracked changes. You should upload this as a separate file labeled ’Manuscript’.

We look forward to receiving your revised manuscript.

Kind regards,

Zhiyuan Ren

Academic Editor

PLOS One

**Journal Requirements:**

1. Please ensure that your manuscript meets PLOS ONE’s style requirements, including those for file naming. The PLOS ONE style templates can be found at

“This study is supported by National Science of Technology Council (NSTC), Taiwan, with Grant no. 112-2221-E-324-010 – and 114-2425-2221-H-028-003-.”

5. Please upload a new copy of Figures 1, 4, 5, 6, 7, 8, 9, 10, 11, and 12 as the detail is not clear. Please follow the link for more information:  https://journals.plos.org/plosone/s/figures

6. Thank you for providing your underlying data as Supporting Information.

We note that the data set contains text or data that is not in English. Please note that PLOS is an English-language publisher, so we require data sets to be provided in English as well. Please upload an English-language version of your data set.

This will also allow us to determine if your data follows PLOS standards per our Data Availability policy here: https://journals.plos.org/plosone/s/data-availability

Reviewers’ comments:

Reviewer’s Responses to Questions

**Comments to the Author**

1. Is the manuscript technically sound, and do the data support the conclusions?

Reviewer #1: Yes

Reviewer #2: Yes

2. Has the statistical analysis been performed appropriately and rigorously? 

Reviewer #1: Yes

Reviewer #2: Yes

3. Have the authors made all data underlying the findings in their manuscript fully available?

Reviewer #1: Yes

Reviewer #2: Yes

4. Is the manuscript presented in an intelligible fashion and written in standard English?

Reviewer #1: Yes

Reviewer #2: Yes

5. Review Comments to the Author

Reviewer #1: 1. Is the AOC method limited to a specific type of NLOS environment or does it work well in different conditions (e.g. with walls, different obstacles)?

2. Why was fuzzy triple std chosen as the best input membership function?

3. Could you elaborate on the number and type of input and output variables of the fuzzy inference system used for area detection?

4. Please provide more details about the layout of the twelve test points used in the experiments. Are these points uniformly distributed in the test area or are they located at specific points with more NLOS challenges?

5. Describe the hardware used in UWB and AoA used in the experiments.

Reviewer #2: 1. Clarify the motivation.

2. Highlight the novelty more explicitly.

3. Provide a short description of the fuzzy membership function types.

4. Picture Quality of all the Figures should be improved.

5. Flowchart representation should be refined for better understanding.

6. Include the recent findings & survey (2024 & 2025).

7. Mathematical notations to be included properly.

8. Briefly indicate why the procedure is divided into two phases.

9. Clarify the role of the moving average filter in both phases.

10. Mention sampling duration or amount of data collected.

11. State how the offset is stored and associated with specific zones.

12. Strengthen the last sentence by clearly stating the outcome.

13. Add an overall summary statement in the conclusion.

14. Add a future work statement.

6. PLOS authors have the option to publish the peer review history of their article (what does this mean?). If published, this will include your full peer review and any attached files.

Reviewer #1: No

Reviewer #2: No

---

## [Author Response · Author response to Decision Letter 1]

3 Feb 2026

Dear Editor,

Thank you for your kind notification. We have clearly responded following requirements:

1. Our manuscript meets PLOS ONE’s style requirements, including those for file naming.

2. All author-generated code is made available without restrictions upon publication of the work.

3. The Funding Information is disclosure by:

“This study is supported by National Science of Technology Council (NSTC), Taiwan, with Grant no. 114-2221-E-324-005 – and 114-2425-2221-H-028-003-.”

4. We have uploaded a new copy of Figures 1, 2, 3, 4, 5, 6, 7, 8, 9, 10, 11, and 12.

5. The data set is provided in English.

6. We cited some specific previously published works and reviewed these referred publications.

We apreiciate the reviewers’ comments. We have clearly responded the comments in the file of "Response to reviewers.docx".

---

## [Decision Letter · Decision Letter 1]

25 Feb 2026

PONE-D-25-27749R1Precision improvement for indoor positioning based on fuzzy inference system with ultra-wideband wireless communicationsPLOS One

Dear Dr. Huang,

Thank you for submitting your manuscript to PLOS ONE. After careful consideration, we feel that it has merit but does not fully meet PLOS ONE’s publication criteria as it currently stands. Therefore, we invite you to submit a revised version of the manuscript that addresses the points raised during the review process.

A letter that responds to each point raised by the academic editor and reviewer(s). You should upload this letter as a separate file labeled ’Response to Reviewers’.A marked-up copy of your manuscript that highlights changes made to the original version. You should upload this as a separate file labeled ’Revised Manuscript with Track Changes’.An unmarked version of your revised paper without tracked changes. You should upload this as a separate file labeled ’Manuscript’.

We look forward to receiving your revised manuscript.

Kind regards,

Zhiyuan Ren

Academic Editor

PLOS One

Journal Requirements:

Reviewers’ comments:

Reviewer’s Responses to Questions

**Comments to the Author**

1. If the authors have adequately addressed your comments raised in a previous round of review and you feel that this manuscript is now acceptable for publication, you may indicate that here to bypass the “Comments to the Author” section, enter your conflict of interest statement in the “Confidential to Editor” section, and submit your "Accept" recommendation.

Reviewer #1: All comments have been addressed

Reviewer #2: All comments have been addressed

2. Is the manuscript technically sound, and do the data support the conclusions?

Reviewer #1: Yes

Reviewer #2: Yes

3. Has the statistical analysis been performed appropriately and rigorously? 

Reviewer #1: Yes

Reviewer #2: Yes

4. Have the authors made all data underlying the findings in their manuscript fully available?

Reviewer #1: Yes

Reviewer #2: Yes

5. Is the manuscript presented in an intelligible fashion and written in standard English?

Reviewer #1: Yes

Reviewer #2: Yes

6. Review Comments to the Author

Reviewer #1: 1. Given that your work focuses on improving accuracy in NLOS environments, how can your system adapt to environmental changes in real time?

2. In the experiments, your test environment is limited to a 600x800 cm space. How do you think the FAOC method will scale in larger or more complex environments (such as multi-story buildings or smart cities)?

3. You have compared your fuzzy inference system with clear area recognition and K-means methods. The question arises as to how your method performs compared to more advanced methods such as machine learning and big data-based models for reducing NLOS errors?

4. In the results, you have shown that AOC was able to reduce the positioning error by an average of 29.38 cm, but in some cases the final error is still around 20 cm. What plans do you have to reduce this error in future versions? Can improvements in signal processing or hardware help to achieve sub-20 cm accuracy?

5. Is your method for correcting offset values stable enough over time and with changing environmental conditions?

6. Is this system currently compatible with systems on the market that use technologies such as Wi-Fi or UWB?

Reviewer #2: (No Response)

7. PLOS authors have the option to publish the peer review history of their article (what does this mean?). If published, this will include your full peer review and any attached files.

Reviewer #1: No

Reviewer #2: No

---

## [Author Response · Author response to Decision Letter 2]

15 Apr 2026

Dear Editor-in-Chief,

Thank you for your kind notification. Please kind find attached file "Response to Reviewers" in which we have clearly responded following requirements:

1. Given that your work focuses on improving accuracy in NLOS environments, how can your system adapt to environmental changes in real time?

2. In the experiments, your test environment is limited to a 600x800 cm space. How do you think the FAOC method will scale in larger or more complex environments (such as multi-story buildings or smart cities)?

3. You have compared your fuzzy inference system with clear area recognition and K-means methods. The question arises as to how your method performs compared to more advanced methods such as machine learning and big data-based models for reducing NLOS errors?

4. In the results, you have shown that AOC was able to reduce the positioning error by an average of 29.38 cm, but in some cases the final error is still around 20 cm. What plans do you have to reduce this error in future versions? Can improvements in signal processing or hardware help to achieve sub-20 cm accuracy?

5. Is your method for correcting offset values stable enough over time and with changing environmental conditions?

6. Is this system currently compatible with systems on the market that use technologies such as Wi-Fi or UWB?

---

## [Decision Letter · Decision Letter 2]

27 Apr 2026

Precision improvement for indoor positioning based on fuzzy inference system with ultra-wideband wireless communications

PONE-D-25-27749R2

Dear Dr. Huang,

We’re pleased to inform you that your manuscript has been judged scientifically suitable for publication and will be formally accepted for publication once it meets all outstanding technical requirements.

An invoice will be generated when your article is formally accepted. Please note, if your institution has a publishing partnership with PLOS and your article meets the relevant criteria, all or part of your publication costs will be covered. Please make sure your user information is up-to-date by logging into Editorial Manager at Editorial Manager® and clicking the ‘Update My Information’ link at the top of the page. For questions related to billing, please contact billing support.

Kind regards,

Zhiyuan Ren

Academic Editor

PLOS One

Additional Editor Comments (optional):

Reviewers’ comments:

Reviewer’s Responses to Questions

**Comments to the Author**

1. If the authors have adequately addressed your comments raised in a previous round of review and you feel that this manuscript is now acceptable for publication, you may indicate that here to bypass the “Comments to the Author” section, enter your conflict of interest statement in the “Confidential to Editor” section, and submit your "Accept" recommendation.

Reviewer #1: (No Response)

Reviewer #2: All comments have been addressed

2. Is the manuscript technically sound, and do the data support the conclusions?

Reviewer #1: Yes

Reviewer #2: Yes

3. Has the statistical analysis been performed appropriately and rigorously? 

Reviewer #1: Yes

Reviewer #2: N/A

4. Have the authors made all data underlying the findings in their manuscript fully available?

Reviewer #1: Yes

Reviewer #2: Yes

5. Is the manuscript presented in an intelligible fashion and written in standard English?

Reviewer #1: Yes

Reviewer #2: Yes

6. Review Comments to the Author

Reviewer #1: (No Response)

Reviewer #2: (No Response)

7. PLOS authors have the option to publish the peer review history of their article (what does this mean?). If published, this will include your full peer review and any attached files.

Reviewer #1: No

Reviewer #2: No

---

## [Editor Report · Acceptance letter]

PONE-D-25-27749R2

PLOS One

Dear Dr. Huang,

I’m pleased to inform you that your manuscript has been deemed suitable for publication in PLOS One. Congratulations! Your manuscript is now being handed over to our production team.

Lastly, if your institution or institutions have a press office, please let them know about your upcoming paper now to help maximize its impact. If they’ll be preparing press materials, please inform our press team within the next 48 hours. Your manuscript will remain under strict press embargo until 2 pm Eastern Time on the date of publication. For more information, please contact onepress@plos.org.

Kind regards,

on behalf of

Professor Zhiyuan Ren

Academic Editor

PLOS One